# Dietary Cholic Acid Exacerbates Liver Fibrosis in NASH Model of Sprague–Dawley Rats Fed a High-Fat and High-Cholesterol Diet

**DOI:** 10.3390/ijms23169268

**Published:** 2022-08-17

**Authors:** Mayuko Ichimura-Shimizu, Shiro Watanabe, Yuka Kashirajima, Ami Nagatomo, Hitomi Wada, Koichi Tsuneyama, Katsuhisa Omagari

**Affiliations:** 1Department of Pathology and Laboratory Medicine, Institute of Biomedical Sciences, Tokushima University Graduate School, 3-18-15 Kuramoto, Tokushima 770-8503, Japan; 2Department of Nutrition Science, Faculty of Nursing and Nutrition, University of Nagasaki, Siebold, Nagasaki 851-2195, Japan; 3Department of Food Science and Nutrition, Nara Women’s University, Kita-Uoya Nishimachi, Nara 630-8506, Japan; 4Institute of Natural Medicine, University of Toyama, 2630 Sugitani, Toyama 930-0194, Japan

**Keywords:** cholic acid, liver fibrosis, nonalcoholic steatohepatitis, high-fat and high-cholesterol diet, Sprague–Dawley rat, bile acid

## Abstract

Background: Recently, we established a novel rodent model of nonalcoholic steatohepatitis (NASH) with advanced fibrosis induced by a high-fat and high-cholesterol (HFC) diet containing cholic acid (CA), which is known to cause hepatotoxicity. The present study aimed to elucidate the direct impact of dietary CA on the progression of NASH induced by feeding the HFC diet. Methods: Nine-week-old male Sprague–Dawley rats were randomly assigned to receive a normal, HFC, or CA-supplemented (0.1%, 0.5% or 2.0%, *w*/*w*) HFC diet for 9 weeks. Results: Histopathological assessment revealed that the supplementation of CA dose-dependently aggravated hepatic steatosis, inflammation, and fibrosis, reaching stage 4 cirrhosis in the 2.0% CA diet group. In contrast, the rats that were fed the HFC diet without any added CA developed mild steatosis and inflammation without fibrosis. The hepatic cholesterol content and mRNA expression involved in inflammatory response and fibrogenesis was higher in a CA dose-dependent manner. The hepatic chenodeoxycholic acid levels were higher in 2.0% CA diet group than in the control, although hepatic levels of total bile acid and CA did not increase dose-dependently with CA intake. Conclusion: Adding CA to the HFC diet altered bile acid metabolism and inflammatory response and triggered the development of fibrosis in the rat liver.

## 1. Introduction

Bile acids are synthesized from cholesterol in the liver and stored in the gall bladder. They are secreted into the intestine as a constituent of bile upon ingestion of a fatty meal, from whence they facilitate digestion and absorption of dietary fats and lipid-soluble vitamins [1,2]. In addition, bile acids regulate lipid, glucose, and energy metabolism as signaling molecules. Despite their physiological importance, bile acids at certain concentrations produce toxicity to membrane components of cells. Elevated intrahepatic bile acid levels cause hepatocellular injury, as seen in cholestatic liver disease [3].

Nonalcoholic fatty liver disease (NAFLD) is a spectrum of liver disorders including nonalcoholic fatty liver (NAFL), nonalcoholic steatohepatitis (NASH), and cirrhosis and is classified according to histological criteria. The prevalence of NAFLD is estimated to reach 30% of adults in developed countries, and more than 10% of cases progress to having hepatic lesions including lobular inflammation, hepatocyte degeneration or “ballooning”, and fibrosis in addition to fat deposition, which are hallmarks of NASH [4]. NAFLD/NASH is frequently accompanied by obesity and is considered to be a hepatic manifestation of metabolic syndrome [5]. Presently, no specific established therapy exists to prevent NASH progression and subsequent cirrhosis, partly due to the lack of good mouse models replicating the pericellular fibrosis characteristics of NASH.

Dietary composition greatly affects the development of metabolic syndrome including NAFLD/NASH [6]. In animal models, there is growing evidence that dietary cholesterol intake and consequent increased hepatic cholesterol are critical factors in the development of hepatic steatosis and steatohepatitis [7,8]. We have previously established a new high-fat/cholesterol/cholic acid (HFCC) diet to make animal models of NASH-related fibrosis mimicking the human pattern of NASH [9,10]. Cholic acid (CA), one of the primary bile acids, is commonly used to promote absorption of excessive amounts of cholesterol, as used in an atherogenic diet [11]. Some of the mice fed HFCC diets supplemented with various concentrations of CA exhibited interesting liver pathologies, such as loss of fatty degeneration and localized severe inflammation, as seen in autoimmune hepatitis in our study. To our knowledge, few studies have been conducted to elucidate the impact of CA on NASH progression, although some studies have examined the effects of CA alone on cholestatic liver diseases [12]. The present study demonstrates the influence of dietary CA under high-fat and cholesterol intake on liver pathology in a NASH rat model.

## 2. Results

In the preliminary study, a high-fat and high-cholesterol (HFC) diet with 0.5% CA, the CA concentration level commonly used in mice, induced the loss of the characteristic histological findings of NASH in the liver of C57BL6/J mice (Appendix A). Therefore, to examine the high-dose and direct effects of CA, we used Sprague–Dawley (SD) rats that were unaffected by endogenous bile acid because rats do not have a gallbladder.

### 2.1. Body Weight, Cumulative Energy Intake, and Relative Organ Weight

The CA concentration in the HFC, low-CA (L-CA), middle-CA (M-CA), and high-CA (H-CA) diet were 0%, 0.1%, 0.5%, and 2.0% (*w*/*w*), respectively. The cumulative energy intake was higher in the HFC, L-CA, M-CA, and H-CA groups than in the control (*p* < 0.001 for all, Table 1), and the H-CA group showed significantly lower cumulative energy intake compared with the HFC and M-CA groups (*p* = 0.006 and 0.011 respectively). Body weight gain was lower in the H-CA group than in the HFC, L-CA, and M-CA groups (*p* = 0.001, *p* < 0.001, and *p* < 0.001, respectively). Food efficacy was lower in the H-CA group than in the other groups. The liver weight/body weight ratios at 18 weeks of age were higher in a dose-dependent manner (*p* = 0.040, 0.002, *p* < 0.001, *p* < 0.001 vs. HFC, L-CA, M-CA, or H-CA group for control group; *p* < 0.001 vs. M-CA or H-CA group for HFC group; *p* = 0.017 and *p* < 0.001 vs. M-CA or H-CA group for L-CA group). Conversely, the ratio of epididymal fat pad weight/body weight at 18 weeks of age was higher in the HFC group than in the control group (*p* = 0.033) and was dose-dependently decreased in all the CA groups compared with the HFC group, reaching statistical significance in the H-CA group (*p* = 0.024, *p* < 0.001, *p* < 0.001, *p* < 0.001 vs. control, HFC, L-CA, and M-CA group for H-CA group, Table 1).

### 2.2. Histopathological Findings of the Livers

Gross livers from rats fed a normal diet for 9 weeks retained dark red coloration (Figure 1). The consumption of the HFC diet resulted in an increase in gross size and yellowish coloration of the livers. These changes were exacerbated by adding CA, and the surfaces of the livers were unsmooth in the H-CA group.

Representative liver images and histological assessments are shown in Figure 1 and Table 2. No obvious findings of hepatic steatosis, inflammation, or fibrosis were seen in any of the rats of the control group. Moderate steatosis (grade 2) was observed in all five rats of the HFC group. In contrast, severe steatosis (grade 3) was observed in most of the rats in the L-CA, M-CA, and H-CA groups. Mild-to-moderate lobular inflammation (grade 1 or 2) without ballooning hepatocytes was seen in the HFC group, whereas severe inflammation (grade 3) was seen in two of five rats (40%), three of six rats (50%), and four of five rats (80%) in the L-CA, M-CA, and H-CA groups, respectively. Ballooning degeneration of hepatocytes was evident in two of six rats (33%) and three of five rats (60%) in the M-CA and H-CA groups, respectively, whereas none of the five L-CA group rats showed obvious hepatocyte ballooning. According to the NAFLD activity score (NAS), three of five rats (60%), five of six rats (83%) and five of five rats (100%) in the L-CA, M-CA and H-CA group, respectively, were diagnosed with NASH (i.e., having a NAS of 5 or greater), whereas none of the 10 rats in the control and HFC groups were diagnosed with NASH (*p* < 0.001). Fibrotic changes were absent in the control and HFC groups, but mild fibrosis (stage 1) was found in one of the five rats (20%) in the L-CA group, and perisinusoidal and portal/periportal or bridging fibrosis (stage 2 or 3) was seen in all six rats in the M-CA group. Stage 2 or 4 fibrosis was seen in all five rats in the H-CA group, and of these, cirrhosis (stage 4) was obvious in three of the five rats (60%).

### 2.3. Serum and Hepatic Biochemical Parameters

As shown in Table 3, the serum total cholesterol (TC) levels were significantly higher in the M-CA and H-CA groups than in the control (*p* = 0.007 and *p* < 0.001, respectively), HFC (*p* = 0.004 and *p* < 0.001, respectively), and L-CA (*p* = 0.006 and *p* < 0.001, respectively) groups. The serum TC levels in the H-CA group were higher than those in the M-CA group (*p* = 0.001). Similarly, serum free cholesterol, aspartate aminotransferase (AST) levels were higher in the M-CA and H-CA groups than in the control (*p* = 0.029 and *p* < 0.001, respectively), HFC (*p* = 0.006 and *p* < 0.001, respectively), and L-CA (*p* = 0.017 and *p* < 0.001, respectively) groups. Serum FC levels in the H-CA group were higher than those in the M-CA group (*p* < 0.001). Serum leptin levels in the HFC group were significantly higher than those in the control group (*p* < 0.001), and these levels in the L-CA, M-CA and H-CA groups were lower than those in the HFC group in a dose-dependent manner (*p* = 0.004, *p* < 0.001 and *p* < 0.001, respectively). Serum AST and alanine aminotransferase (ALT) levels tended to be higher in the HFC group than in the control group, and they were dose-dependently increased in all the CA groups, although the differences reached statistical significance only in the H-CA group versus control (AST; *p* = 0.049, ALT; *p* = 0.039). Levels of serum glucose and insulin tended to be higher in the HFC groups than in the control group, and these levels tended to be lower in all the CA groups than in the HFC group, although those levels did not significantly differ among groups. Figure 2 shows a stepwise increase in hepatic TC content from control to the HFC and all CA groups (control, HFC, L-CA, M-CA, and H-CA group: 5 ± 3, 72 ± 4, 109 ± 6, 125 ± 5, and 151 ± 83 mg/g liver; *p* < 0.001 vs. control for all; *p* = 0.005 or *p* < 0.001 vs. M-CA or H-CA for HFC; *p* = 0.043 vs. L-CA for H-CA).

### 2.4. Hepatic mRNA Expression

To determine the impact of dietary CA on the features of NASH, the hepatic expression profiles of genes responsible for inflammatory response, fibrogenesis, and lipid and bile acid metabolism were analyzed (Figure 3). The mRNA levels of monocyte chemotactic protein-1 (*Mcp-1*) were increased in a dose-dependent manner relative to CA intake. The expression levels of procollagen type I, alpha 1, and the profibrotic marker transforming growth factor *(Tgf)*-β1 were increased in a dose-dependent manner in the CA groups, consistent with the observed histological fibrotic changes in the liver. Indeed, the mRNA expression of Mcp-1 and Tgf-β1 were positively and strongly correlated with NAS and fibrosis stage (Figure 4).

The mRNA levels of fatty acid synthase were lower in the HFC and CA groups than in the control group. The mRNA levels of carnitine palmitoyltransferase-1a, a rate-limiting fatty acid transporter involved in β oxidation, were significantly lower in the M-CA group than in the control group. The mRNA levels of microsomal triglyceride transfer protein, the rate-limiting step in the synthesis and excretion of very-low-density lipoprotein from the liver, were lower in the HFC, M-CA, and H-CA groups than in the control group.

### 2.5. Bile Acid Profile in the Liver

To investigate the possibility that some hydrophobic bile acids drive hepatocellular damage and fibrosis, the bile acid profile in the liver was analyzed. Hepatic CA content was lower in the L-CA group than in the control group and was dose-dependently increased in CA-supplemented groups, although there was no significant difference between the H-CA and control groups (Table 4). Hepatic chenodeoxycholic acid (CDCA) content was higher in the H-CA group than in the control group. Rodent-specific α-muricholic acid (MCA) and β-MCA levels synthesized from CDCA did not show the same tendency as CDCA levels. The levels of ω-MCA and hyodeoxycholic acid, secondary bile acids, were lower in a dose-dependent manner. Hepatic total bile acid levels did not differ significantly among groups. Hepatic CDCA levels positively and weakly correlated with NAS and fibrosis stage, whereas hepatic total cholesterol levels were positively and strongly correlated with them (Figure 4).

## 3. Discussion

To explain the pathogenesis of NASH, the “two-hit” theory has been widely accepted; namely, after a first “hit” (steatosis), another “hit” such as oxidative stress is needed for development of NASH [13]. Since then, a growing body of evidence has suggested that multiple factors may take place in parallel, rather than consecutively, as the second “hit(s)”. These factors may include pro-inflammatory cytokines, gut-derived endotoxins, oxidative stress, endoplasmic reticulum stress, lipotoxicity, and activation of intracellular signaling pathways [14]. However, the overall mechanisms involved in the progression from steatosis to NASH are more complex and remain largely unknown.

In our previous study, the rats fed the HFC diet containing 2% CA developed steatohepatitis with obvious fibrosis, whereas those fed the high-fat diet containing 2% CA and no cholesterol developed mild steatosis and inflammation without fibrotic change. This result suggested that cholesterol is an important dietary component that can induce the second “hit”, especially for fibrosis [10]. In contrast, the HFC diet without CA in the present study induced steatosis and inflammation, but not fibrosis. The addition of CA to the HFC diet led to the development of fibrosis that exhibited dose-dependent severity in conjunction with increased hepatic gene expression involved in fibrogenesis. Taken together, histopathological aggravation of the liver, especially fibrosis, can be affected strongly by dietary CA rather than cholesterol under a high-fat loaded condition. These results are in agreement with a previous genome-based study that indicated that the activation of collagen gene family members involved in fibrogenesis is dependent on the presence of CA in the diet, whereas inflammatory genes are induced by cholesterol [15].

Unlike humans, rodents have fewer hydrophobic and cytotoxic MCAs, such as CDCA and DCA, and instead have more hydrophilic MCAs. High levels of hepatoprotective bile acids may explain why rodents rarely develop liver fibrosis [16]. To solve the problem of species differences, several animal models have been reported using genetically modified mice lacking the enzymes involved in MCA synthesis to achieve human bile acid composition and cats that cannot originally synthesize MCA [16,17]. However, rodent fibrosis models, the most common and inexpensive laboratory animal, are essential for the development of therapeutic agents for NASH fibrosis. A diet enriched in fat and cholesterol is well known to cause hepatic steatosis and inflammation in animal models [18]. However, there is little evidence to indicate that fat and cholesterol are sufficient by themselves to result in fibrosis, the hallmark that differentiates progressive NASH from NAFL [19,20]. The results of the present study showed that not only cholesterol, but also CA, may be required to produce progressive NASH including fibrosis in rats in a practical period of time, although supplemental CA does not resemble a dietary constituent in humans.

Bile acids, including CA, disrupt cell membranes through their detergent action on lipid components and can cause oxidative stress and activation of Kupffer cells, eventually producing hepatocyte necrosis and apoptosis [21]. In the present study, liver CA levels did not change in a CA-dependent manner. The lower amount of hepatic CA levels in the L-CA group might be due to the negative feedback of exogenous CA on bile acid synthesis. Excess CA administration can increase the systemic CA pool, which may have resulted in higher levels of hepatic CA in the M-CA and H-CA groups. These results indicate that fibrosis in the rat model cannot be attributed solely to the cytotoxicity of the administered CA. It is worth noting that secondary bile acids including HFCA and ω-MCA, which have few hydroxyl groups and are strongly cytotoxic, were not associated with fibrosis severity. Liver CDCA, which was positively correlated with fibrosis stage in the present study, has been reported to have an activating effect on the NLRP4 inflammasome in vitro [22]. Kwan et al. reported that the levels of plasma conjugated CDCA is strongly correlated with liver fibrosis in humans [23]. Collectively, CDCA may be associated with the pathogenesis of fibrosis in this model. Given the species differences in bile acid profiles, there is a limitation in referring only to CDCA level as a driving factor for fibrosis pathology [17]. However, since few studies have clarified the characteristics of bile acid profile in CA loading under high-fat and cholesterol in animal models, this study may be useful for understanding the background pathophysiology of HFCC-induced NASH rat model.

Oral administration of CA has been reported to attenuate hepatic steatosis in animal models fed with a high-fat or a choline-deficient diet [24,25]. Gabbi et al. also demonstrated that increased hepatic bile acid levels resulting from bile duct ligation improved fatty liver [25]. This effect results from the promotion of lipid metabolism and energy expenditure partly mediated by farnesoid X receptor and G-protein-coupled receptor, the endogenous receptors for bile acids [24]. In fact, the HFC diet supplemented with 2.0% CA not only exacerbated liver pathology with stage 4 fibrosis but also induced a decrease in visceral fat mass and blood glucose levels that did not resemble NASH in the context of metabolic syndrome. Moreover, cholesterol is known to act synergistically with triglyceride to increase body weight [26]. The doses of 0.1–0.5% CA might act antagonistically to the 1.25% cholesterol in the diet and thus, did not cause weight loss in the present study. These results point to the importance of setting appropriate CA doses in making animal models of NASH, which require animals to display a variety of phenotypes, including obesity, insulin resistance, and fatty liver.

In conclusion, dietary CA can be an important determinant for progression to hepatic steatosis, inflammation, and especially fibrosis in HFC diet-induced NASH. CA and cholesterol, and probably the effect of the interaction between them, can be a second “hit” and facilitate development of NASH and liver cirrhosis, which require a long time period to develop in animal models. Understanding liver pathology is sometimes complicated because of the overlap between different diseases such as NASH and autoimmune hepatitis, but developing an animal model that substantially reflects the human pathology will help us understand the pathogenesis of NASH.

## 4. Materials and Methods

### 4.1. Animals and Experimental Design

Eight-week-old male SD rats were purchased from Japan SLC (Hamamatsu, Japan) and housed individually in a temperature- and humidity-controlled room with a 12-h light/dark cycle. After 1 week of acclimation with standard rodent chow (MF; Oriental Yeast, Tokyo, Japan) and water ad libitum, the rats were randomly divided into five groups: Control group (*n* = 5), fed standard rodent chow (MF) as the normal diet for 9 weeks; HFC group (*n* = 5), fed a HFC diet for 9 weeks; L-CA group (*n* = 5), fed a HFC diet supplemented with 0.1% (*w*/*w*) sodium cholate for 9 weeks; M-CA group (*n* = 6), fed a HFC diet supplemented with 0.5% (*w*/*w*) sodium cholate for 9 weeks; and H-CA group (*n* = 5), fed a HFC diet supplemented with 2.0% (*w*/*w*) sodium cholate for 9 weeks. All HFC diets were prepared by mixing the MF with 30% (*w*/*w*) palm oil and 1.25% (*w*/*w*) cholesterol. Daily energy intake and body weight were monitored throughout the study.

At 18 weeks of age, the rats were fasted for 8 h and sacrificed under anesthesia with pentobarbital sodium. Organs were harvested and blood was collected from the inferior vena cava. Liver tissues were either put into 10% neutral buffered formalin or snap frozen in liquid nitrogen and stored at −80 °C. All procedures performed on the animals were approved by the Animal Use Committee of the University of Nagasaki, and the animals were maintained in accordance with the guidelines for the care and use of laboratory animals, University of Nagasaki.

### 4.2. Histopathological Assessment of the Livers

Liver tissues stored in 10% neutral buffered formalin were embedded in paraffin, sectioned at 4 μm, and stained with hematoxylin and eosin (HE). Histological steatosis, lobular inflammation, and hepatocyte ballooning were assessed semi-quantitatively to determine the NAFLD activity score (NAS) according to the criteria proposed by Kleiner et al. [27]. The final NAS values ranged from 0 to 8. NAS scores ≥5 and ≤2 were considered diagnostic and not diagnostic, respectively, for steatohepatitis. Liver fibrosis was also assessed by azan staining. All histopathological analyses were performed by a pathologist (K.T.) who was blinded to the study.

### 4.3. Serum and Tissue Biochemical Analysis

Hepatic lipids were extracted as described previously [10]. Serum and/or tissue levels of triglyceride, TC, free cholesterol, glucose, AST, and ALT were determined by Triglyceride E test Wako, Cholesterol E test Wako, Free cholesterol E test Wako, Glucose C II test Wako, and Transaminase C II test Wako (Wako Pure Chemical Industries, Osaka, Japan). Insulin and leptin levels were measured using a rat insulin enzyme-linked immunosorbent assay (ELISA) kit and a Leptin ELISA kit (Morinaga Institute of Biological Science, Yokohama, Japan).

### 4.4. mRNA Quantification by Real-Time PCR

Total RNA from the liver was isolated using RNAiso Plus (Takara Bio, Otsu, Japan) according to the manufacturer’s instructions. RNA was reverse-transcribed to cDNA templates using a commercial kit (PrimeScript RT Master Mix, Takara Bio). Real-time PCR analysis was performed as described previously [10]. Specific primers were designed using the primer designing tool Primer-BLAST (National Center for Biotechnology Information, Bethesda, MD, USA) and were synthesized by Greiner Bio-One Japan (Tokyo, Japan; Table 5). The quantity of mRNA was normalized by glyceraldehyde 3-phosphate dehydrogenase. All data are expressed as fold changes over expression in the control group.

### 4.5. Determination of Bile Acid Levels in the Liver

Protocols for the extraction of bile acids from frozen liver tissues and their determination by liquid chromatography–mass spectrometry (LC–MS) were identical to those described in our previous report [28].

### 4.6. Statistical Analysis

Data are presented as means ± standard error (SE). Differences between groups were analyzed using one-way analysis of variance and the Sheffe post hoc test. Correlations between two variables were determined by Spearman’s rank correlation coefficient. A P value of less than 0.05 was considered to be statistically significant. Statistical analyses were performed with IBM SPSS statistics software, version 24.0 for Windows (IBM Corp., Somers, NY, USA).

## Figures and Tables

**Figure 1 ijms-23-09268-f001:**
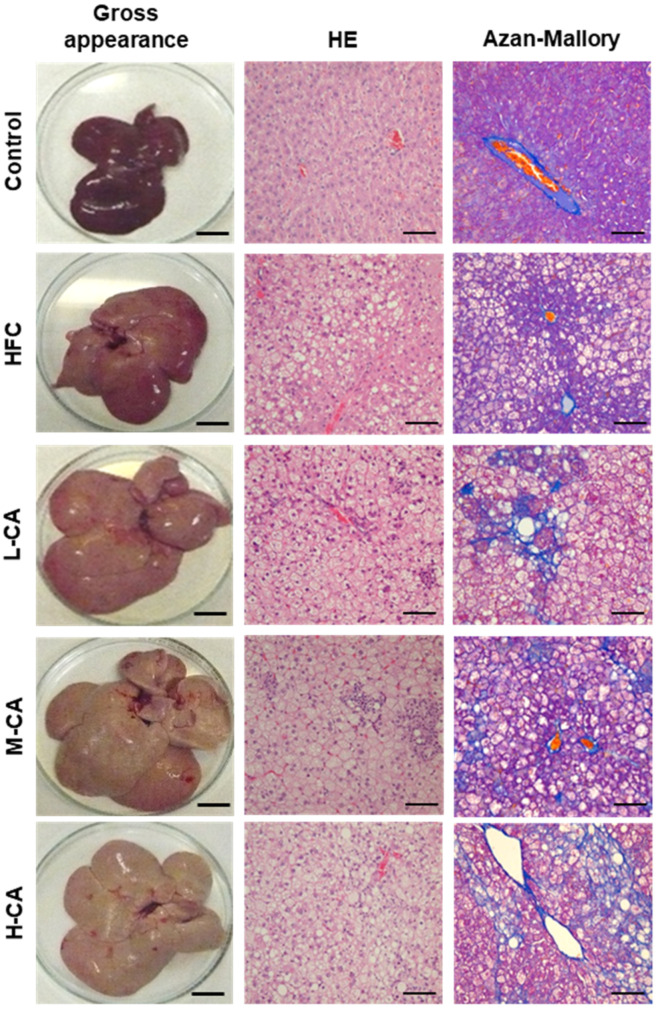
Representative liver morphology and histopathology in the control, HFC and L-CA, M-CA and H-CA groups. Gross appearance of liver surface; scale bars = 4 cm. Hematoxylin and eosin (HE)- and Azan-Mallory stained section; scale bars = 100 μm. HFC, high-fat and high-cholesterol; L-CA, M-CA and H-CA, the HFC diet with 0.1%, 0.5% and 2.0% sodium cholate (*w*/*w*), respectively.

**Figure 2 ijms-23-09268-f002:**
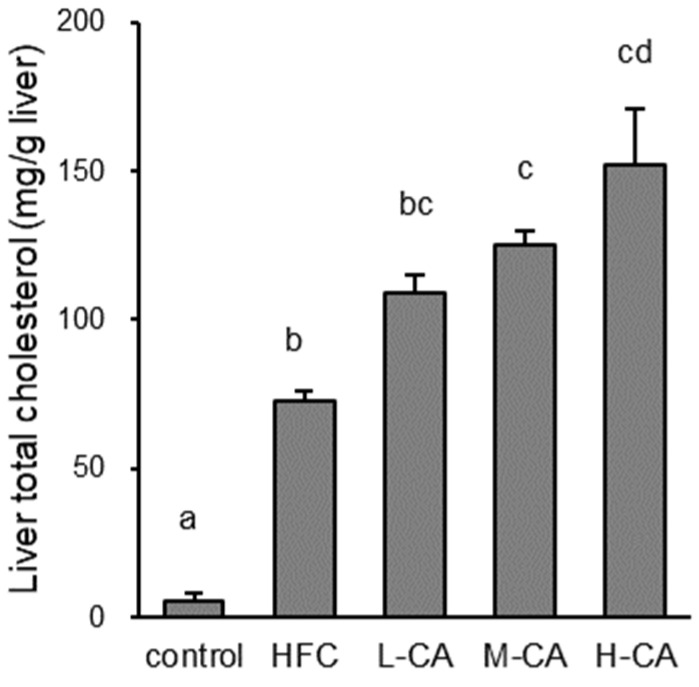
Cholesterol content of the liver in the control, HFC, and L-CA, M-CA and H-CA groups. Values are expressed as means ± SE. *n* = 5–6/group. ^a,b,c,d^ values not sharing the same lowercase letter are significantly different among groups (*p* < 0.05). HFC, high-fat and high-cholesterol; L-CA, M-CA and H-CA, the HFC diet with 0.1%, 0.5% and 2.0% sodium cholate (*w*/*w*), respectively.

**Figure 3 ijms-23-09268-f003:**
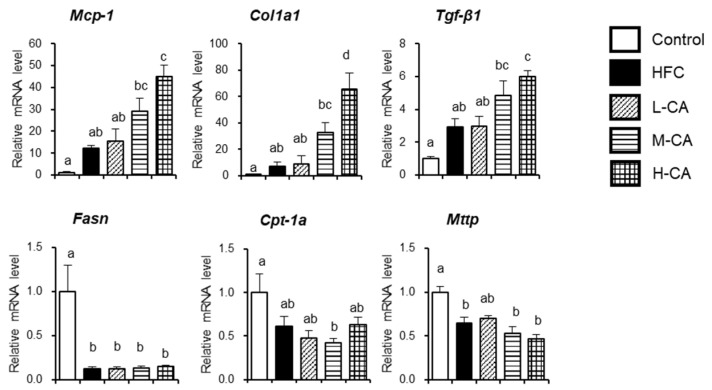
Hepatic gene expression involved in inflammatory response, fibrogenesis, and lipid metabolism in the control, HFC and L-CA, M-CA and H-CA groups. The levels of mRNA were expressed relative to the control group. Values are expressed as means ± SE; *n* = 5–6/group. ^a,b,c,d^ values not sharing the same lowercase letter are significantly different among groups (*p* < 0.05). HFC, high-fat and high-cholesterol; L-CA, M-CA and H-CA, the HFC diet with 0.1%, 0.5% and 2.0% sodium cholate (*w*/*w*), respectively; Col1a1, procollagen type I, alpha 1; Cpt-1a, carnitine palmitoyltransferase-1a; Fasn, fatty acid synthase; Mcp-1, monocyte chemotactic protein-1; Tgf-β1, profibrotic marker transforming growth factor-β1.

**Figure 4 ijms-23-09268-f004:**
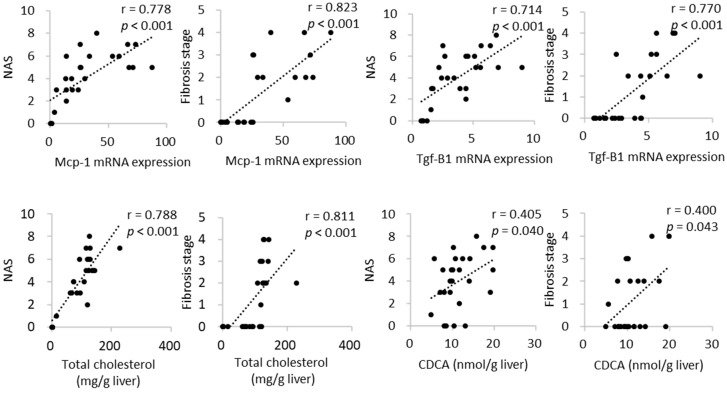
Correlation between histopathological assessment and mRNA expression and content of cholesterol and CDCA in the liver. CDCA, chenodeoxycholic acid; NAS, nonalcoholic fatty liver disease activity score; Mcp-1, monocyte chemotactic protein-1; Tgf-β1, profibrotic marker transforming growth factor-β1.

**Table 1 ijms-23-09268-t001:** Cumulative energy intake, body weight, and relative organ weights in Sprague–Dawley rats fed the normal, HFC and CA-supplemented HFC diet for 9 weeks.

Group	Control	HFC	L-CA	M-CA	H-CA
Cumulative energy intake (kcal)	5711 ± 109 ^a^	7555 ± 141 ^b^	7369 ± 169 ^b,c^	7481 ± 102 ^b^	6832 ± 98 ^c^
Final body weight (g)	525 ± 11 ^a^	574 ± 13 ^a^	584 ± 19 ^a^	574 ± 18 ^a^	480 ± 12 ^b^
Body weight gain (g)	202 ± 8 ^a^	252 ± 10 ^a^	259 ± 17 ^a^	254 ± 16 ^a^	159 ± 13 ^b^
Food efficacy (g/kcal)	0.035 ± 0.001 ^a^	0.033 ± 0.001 ^a^	0.035 ± 0.002 ^a^	0.034 ± 0.002 ^a^	0.023 ± 0.002 ^b^
Liver weight (g)	17.7 ± 1.5 ^a^	28.5 ± 1.6 ^b^	32.7 ± 2.7 ^b^	42.0 ± 3.3 ^c^	42.2 ± 2.3 ^c^
Liver weight/body weight (%)	3.4 ± 0.2 ^a^	5.0 ± 0.2 ^b^	5.6 ± 0.4 ^b^	7.3 ± 0.5 ^c^	8.8 ± 0.2 ^c^
Epididymal fat pad weight/body weight (%)					
	2.1 ± 0.1 ^a^	2.9 ± 0.2 ^b^	2.4 ± 0.2 ^a,b^	2.2 ± 0.3 ^a,b^	1.0 ± 0.2 ^c^

Values are expressed as means ± SE. *n* = 5–6/group. ^a,b,c^ Values not sharing the same lowercase letter in a row are significantly different among groups (*p* < 0.05). Food efficacy was calculated by body weight gain (g)/cumulative energy intake (kcal). HFC, high-fat, and high-cholesterol; L-CA, M-CA, and H-CA, the HFC diet with 0.1%, 0.5% and 2.0% sodium cholate (*w*/*w*), respectively.

**Table 2 ijms-23-09268-t002:** Histopathological assessment of the liver in Sprague–Dawley rats fed the normal, HFC or CA-supplemented HFC diet for 9 weeks.

Group/Rat No.	Steatosis	Lobular Inflammation	Hepatocyte Ballooning	NAFLD Activity Score	Fibrosis
Control-1	0	0	0	0	0
Control-2	1	0	0	1	0
Control-3	0	0	0	0	0
Control-4	0	0	0	0	0
Control-5	0	0	0	0	0
HFC-1	2	2	0	4	0
HFC-2	2	2	0	4	0
HFC-3	2	1	0	3	0
HFC-4	2	1	0	3	0
HFC-5	2	1	0	3	0
L-CA-1	3	2	0	5	0
L-CA-2	1	1	0	2	0
L-CA-3	3	3	0	6	0
L-CA-4	3	3	0	6	1
L-CA-5	2	1	0	3	0
M-CA-1	3	2	0	5	2
M-CA-2	3	1	1	5	3
M-CA -3	2	3	0	5	3
M-CA-4	3	3	1	7	3
M-CA-5	3	1	0	4	2
M-CA-6	3	3	0	6	2
H-CA-1	3	2	0	5	4
H-CA -2	3	3	0	6	2
H-CA-3	2	3	2	7	2
H-CA-4	3	3	2	8	4
H-CA-5	3	3	1	7	4

HFC, high-fat and high-cholesterol; L-CA, M-CA and H-CA, the HFC diet with 0.1%, 0.5% and 2.0% sodium cholate (*w*/*w*), respectively. NAS, nonalcoholic fatty liver disease activity score.

**Table 3 ijms-23-09268-t003:** Serum biochemical profile in Sprague–Dawley rats fed the normal, HFC, and CA-supplemented HFC diet for 9 weeks.

Group	Control	HFC	L-CA	M-CA	H-CA
Triglyceride (mg/dL)	83 ± 20	70 ± 12	69 ± 6	76 ± 10	49 ± 7
Total cholesterol (mg/dL)	38 ± 2 ^a^	36 ± 3 ^a^	38 ± 5 ^a^	81 ± 6 ^b^	132 ± 16 ^c^
Free cholesterol (mg/dL)	16 ± 1 ^a^	12 ± 1 ^a^	13 ± 2 ^a^	30 ± 4 ^b^	63 ± 5 ^c^
Glucose (mg/dL)	144 ± 2	219 ± 41	196 ± 55	155 ± 11	134 ± 6
Insulin (ng/mL)	5.6 ± 1.1	8.7 ± 1.5	6.2 ± 0.6	6.0 ± 1.5	5.7 ± 1.5
Leptin (ng/mL)	8.1 ± 0.6 ^a^	15.0 ± 1.3 ^b^	9.7 ± 1.0 ^a,c^	5.9 ± 0.9 ^a,d^	2.6 ± 0.3 ^d^
AST (IU/L)	43 ± 15 ^a^	67 ± 7 ^a^	74 ± 7 ^a^	81 ± 32 ^a^	146 ± 32 ^b^
ALT (IU/L)	11 ± 4 ^a^	24 ± 4 ^a^	24 ± 1 ^a^	31 ± 9 ^a^	53 ± 11 ^b^

Values are expressed as means ± SE. *n* = 5–6/group. ^a,b,c,d^ values not sharing the same lowercase letter in a row are significantly different among groups (*p* < 0.05). ALT, alanine aminotransferase; AST, aspartate aminotransferase; HFC, high-fat and high-cholesterol; L-CA, M-CA and H-CA, the HFC diet with 0.1%, 0.5% and 2.0% sodium cholate (*w*/*w*), respectively.

**Table 4 ijms-23-09268-t004:** Hepatic bile acids profile in Sprague–Dawley rats fed the normal, HFC, and CA-supplemented HFC diet for 9 weeks.

Group	Control	HFC	L-CA	M-CA	H-CA
ω-MCA (nmol/g liver)	3.91 ± 0.92 ^a^	1.68 ± 0.33 ^a,b^	1.07 ± 0.32 ^b^	0.99 ± 0.49 ^b^	0.53 ± 0.13 ^b^
α-MCA (nmol/g liver)	6.52 ± 1.72	8.04 ± 2.11	6.00 ± 1.92	6.25 ± 0.77	7.90 ± 1.09
β-MCA (nmol/g liver)	6.52 ± 1.72	47.23 ± 11.10	21.57 ± 4.93	20.59 ± 5.05	17.65 ± 0.93
CA (nmol/g liver)	74.99 ± 13.91 ^a,c^	44.13 ± 11.13 ^a,b,c^	27.23 ± 8.26 ^b^	33.55 ± 4.16 ^a,b^	92.80 ± 7.06 ^a,c^
UDCA (nmol/g liver)	2.40 ± 0.46	3.83 ± 1.34	2.98 ± 0.76	3.11 ± 0.61	2.81 ± 0.40
HDCA (nmol/g liver)	0.75 ± 0.12 ^a^	0.21 ± 0.03 ^b^	0.11 ± 0.04 ^b^	0.11 ± 0.06 ^b^	0.12 ± 0.04 ^b^
CDCA (nmol/g liver)	8.95 ± 1.34 ^a^	10.97 ± 2.08 ^a,b^	10.23 ± 1.50 ^a,b^	10.43 ± 0.83 ^a,b^	17.04 ± 1.36 ^b^
DCA (nmol/g liver)	18.40 ± 6.01	19.61 ± 5.10	12.94 ± 2.77	27.53 ± 7.41	14.55 ± 8.80
LCA (nmol/g liver)	0.23 ± 0.08	0.58 ± 0.21	0.11 ± 0.05	0.27 ± 0.09	0.11 ± 0.07
Total bile acids (nmol/g liver)	138.55 ± 23.23	136.28 ± 30.33	82.24 ± 16.76	102.83 ± 14.95	153.52 ± 16.34

Values are expressed as means ± SE. *n* = 5–6/group. ^a,b,c^ values not sharing the same lowercase letter in a row are significantly different among groups (*p* < 0.05). CA, cholic acid; CDCA, chenodeoxycholic acid; DCA; deoxycholic acid; HDCA, hyodeoxycholic acid; LCA, litocholic acid; MCA, muricholic acid; UDCA, urusodeoxycholic acid; HFC, high-fat and high-cholesterol; L-CA, M-CA and H-CA, the HFC diet with 0.1%, 0.5% and 2.0% sodium cholate (*w*/*w*), respectively.

**Table 5 ijms-23-09268-t005:** Primer sets for real-time PCR analysis.

	Forward Sequence	Reverse Sequence
*Col1a1*	GCGTAGCCTACATGGACCAA	AAGTTCCGGTGTGACTCGTG
*Cpt-1a*	ATCGACCGCCATCTCTTCTG	CCATGGCTCAGACAATACCTCC
*Fasn*	CAACATTGACGCCAGTTCCG	TTCGAGCCAGTGTCTTCCAC
*Gapdh*	GGCACAGTCAAGGCTGAGAATG	ATGGTGGTGAAGACGCCAGTA
*Mcp-1*	GCAGTTAATGCCCCACTCAC	TTGAGCTTGGTGACAAATACTACAG
*Mttp*	CAAGCTCAAGGCAGTGGTTG	AGCAGGTACATCGTGGTGTC
*Tgf-β1*	CTTTGTACAACAGCACCCGC	TAGATTGCGTTGTTGCGGTC

Col1a1, procollagen type I, alpha 1; Cpt-1a, carnitine palmitoyltransferase-1a; Fasn, fatty acid synthase; Gapdh, glyceraldehyde 3-phosphate dehydrogenase; Mcp-1, monocyte chemotactic protein-1; Tgf-β1, profibrotic marker transforming growth factor-β1.

## Data Availability

Not applicable.

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
