# Peer review of "Dietary Cholic Acid Exacerbates Liver Fibrosis in NASH Model of Sprague–Dawley Rats Fed a High-Fat and High-Cholesterol Diet"

_ijms, 2022, doi:10.3390/ijms23169268_

Round 1
Reviewer 1 Report
In this study the Authors aimed to elucidate the direct impact of dietary cholic acid (CA) on the progression of NASH induced by feeding the HFC diet. in a rat model. Rats were randomly assigned to receive a normal, HFC, or CA-supplemented (0.1%, 0.5% or 2.0%, 23 w/w) HFC diet for 9 weeks. They found that histopathological assessment revealed that the supplementation of CA dose-dependently aggravated hepatic steatosis, inflammation and fibrosis, reaching stage 4 cirrhosis in the 25 2.0% CA diet group. In contrast, the rats fed the HFC diet without any added CA developed mild steatosis and inflammation without fibrosis. Hepatic cholesterol content and mRNA expression involved in inflammatory response and fibrogenesis was higher in a CA dose-dependent manner. Hepatic chenodeoxycholic acid levels were higher in 2.0% CA diet group than in control, although hepatic levels of total bile acid and CA were not increase in dose-dependently with CA intake.
They concluded that adding CA to the HFC diet altered bile acid metabolism and inflammatory response and triggered development of fibrosis in the rat liver.
The study is clearly presented and well discussed. I only suggest minor English language and style changes.
Author Response
Thank you very much for your positive evaluation. As the referee 1 suggested, there were some linguistic and grammatical errors. Now we’ve corrected them.
Reviewer 2 Report
Major comments:
1. Do you have any explanation for the dramatic decrease in body weight but specifically the increase in liver weight in the H-CA group in Table 1?
2. Do you have any explanation for the dramatic decrease of CA in the L-CA group particularly CA then increases by dose-dependent when fed by additional CA in Table 4?
3. Please also investigate more relationships between fibrosis/NAS and total cholesterol, MCP-1, and TGF-beta1 mRNA expression in Fig 4.
Author Response
Thank you very much for your valuable comments. Your comments have been very helpful in allowing us to revise our manuscript. Now we revised our manuscript accordingly.
Point 1: Do you have any explanation for the dramatic decrease in body weight but specifically the increase in liver weight in the H-CA group in Table 1?
Response 1: Factors affecting the liver and body weight may include exogenous cholesterol and CA.
First, liver weight itself was included in the results because the marked increase in liver/body weight ratio in the H-CA group was thought to be a relative result due to the decrease in body weight (Table 1). Cholesterol has been reported to act synergistically with triglyceride to increase body weight and liver fat. The increase in liver weights in the HFC, L-CA, and M-CA groups indicates that administration of 0%-0.5% CA increases cholesterol absorption in a dose-dependent manner. In addition, there is no significant difference in liver weight of H-CA group (2.0% CA feeding) compared to M-CA group (0.5% CA feeding) suggesting that the 0.5% CA dose is sufficient for absorption of 1.25% cholesterol in the diet.
Besides promoting lipid absorption, CA acts as a ligand for y farnesoid X receptor and G-protein-coupled receptor to increase energy expenditure and decrease adiposity. Conversely, cholesterol acts to increase body weight. Given these opposing effects, the following explanation can be offered for the results of body weight in this study: In the L-CA and M-CA groups, cholesterol-dominant effects did not reduce body weight, even when CA was fed at the same time. In contrast, the CA-dominant effect caused weight loss in the H-CA group. In this context, we added the text in the discussion (lines 322-330).
Point 2: Do you have any explanation for the dramatic decrease of CA in the L-CA group particularly CA then increases by dose-dependent when fed by additional CA in Table 4?
Response 2: Bile acid metabolism is feedback regulated via farnesoid X receptor. One plausible explanation for the decreased CA levels in the L-CA group can be that exogenous CA induced a negative feedback on bile acid synthesis in the liver. Also, excess CA administration increase the systemic CA pool, which may have resulted in higher hepatic CA levels in the M-CA and H-CA groups. In this context, we added the text in the discussion (lines296 -301).
Point 3: Please also investigate more relationships between fibrosis/NAS and total cholesterol, MCP-1, and TGF-beta1 mRNA expression in Fig 4.
Response 3: Correlation between fibrosis/NAS and total cholesterol, MCP-1, and TGF-beta1 mRNA expression were analyzed (Fig.4, lines 192-194 and 226-228). The mRNA expression of Mcp-1 and Tgf-β1 were positively and strongly correlated with NAS and fibrosis stage. Hepatic CDCA levels positively and weakly correlated with NAS and fibrosis stage, while hepatic total cholesterol levels was positively and strongly correlated with them, suggesting that cholesterol accumulation is strongly associated with NASH and fibrosis progression, although the causal relationship is unclear.
Round 2
Reviewer 2 Report
I have no more questions.